# Resizing of the Gastric Pouch for Weight Regain after Laparoscopic Roux-en-Y Gastric Bypass and One-Anastomosis Gastric Bypass: Is It a Valid Option?

**DOI:** 10.3390/jcm11216238

**Published:** 2022-10-22

**Authors:** Silvia Ferro, Viola Zulian, Massimiliano De Palma, Andrea Sartori, Anamaria Andreica, Marius Nedelcu, Sergio Carandina

**Affiliations:** 1Clinica Madonna della Salute, Department of Digestive and Bariatric Surgery, 45014 Porto Viro, Italy; 2Clinica Chirurgica Ospedale Sant’Anna, Università Degli Studi di Ferrara, 44100 Ferrara, Italy; 3ELSAN, Clinique Saint Michel, Centre Chirurgical de l’Obésité (CCO), Clinique Saint Michel, 4, Place du 4 Septembre, 83100 Toulon, France

**Keywords:** gastric bypass, one-anastomosis gastric bypass, weight regain, pouch resizing, anastomosis recalibration

## Abstract

Introduction: The laparoscopic resizing of the gastric pouch (LPR) has recently been proposed as a revisional technique in the case of weight regain (WR) after gastric bypass procedures. The aim of this study was to report our experience with LPR for WR. Materials and Methods: All patients with WR ≥ 25% after gastric bypass and with a dilated gastric pouch and/or gastrojejunal anastomosis who underwent LPR between January 2017 and January 2022 were retrospectively reviewed. From a radiological point of view, a gastric pouch was considered dilated when its volume was calculated at >80 cm^3^ for LRYGB and >200 cm^3^ for OAGB upon a 3D-CT scan. The endoscopic criterion considered both the diameter of the gastrojejunal anastomosis and the gastric pouch volume. All anastomoses > 20 mm for LRYGB and >40 mm for OAGB were considered dilated, while a gastric pouch was considered endoscopically dilated when the retrovision maneuver with the gastroscope was easily performed. These selection criteria were arbitrarily established on the basis of both our personal experience and literature data. Results: Twenty-three patients had LPR after a Roux-en-Y gastric bypass or one-anastomosis gastric bypass. The mean BMI at LPR was 36.3 ± 4.7 kg/m^2^. All patients underwent LPR, while the resizing of the GJA was also performed in 3/23 (13%) cases, and hiatoplasty was associated with the resizing of the pouch in 6/23 cases (26.1%). The mean BMI at the last follow-up was 29.3 ± 5.8 kg/m^2^. The difference between the BMI before resizing and the BMI at the last follow-up visit was statistically significant (*p* = 0.00005). The mean %TWL at 24.2 ± 16.1 months was 19.6 ± 9%. Comorbidities had an overall resolution and/or improvement rate of 47%. The mean operative time was 71.7 ± 21.9 min. The conversion rate was nil. Postoperative complications occurred in two cases (8.7%). Conclusions: In our series, LPR for WR showed good results in weight loss and in improvement/resolution of comorbidities, with an acceptable complication rate and operative time. Only further studies with a greater cohort of patients and a longer postoperative follow-up will be able to highlight the long-term benefits of this technique.

## 1. Introduction

Bariatric surgery is currently the most effective approach for morbid obesity, with excellent long-term results [1]. In particular, laparoscopic Roux-en-Y gastric bypass (LRYGB) and one-anastomosis gastric bypass (OAGB) are the second- and third-most performed bariatric procedures in the world, respectively [2]. However, despite the undisputed benefits of these procedures, a certain number of patients develop postoperative weight regain (WR) [3]. WR represents a thorny problem, leading to the recovery of obesity-related pathologies, worsening the quality of life, and the need for corrective reinterventions with high intra- and postoperative risks. WR, given its multifactorial nature, can occur at different rates after any bariatric surgical technique [4,5]. In addition to behavioral, nutritional, and psychological disorders, anatomical anomalies have also been accepted as a cause of WR or insufficient weight loss after bariatric surgery. The anatomical causes suggested in the literature after gastric bypass are the dilation of the gastric pouch and/or gastrojejunal anastomosis (GJA). In these cases, treatment options are limited, and literature data are scarce [6]. During the last few years, returning to normal anatomy with a sleeve gastrectomy (LSG) and the placement of an adjustable gastric band at the gastric pouch level in order to restore the restrictive component of the bypass have been proposed, with debatable results [7,8,9,10,11,12,13]. A third option proposed with the same aim is the laparoscopic resizing of the gastric pouch (LPR), with or without the narrowing of the GJA. The purpose of the present study was to analyze our experience with LPR in patients that experienced WR and the recovery of obesity-related comorbidities after gastric bypass.

## 2. Materials and Methods

### 2.1. Study Population

All patients who underwent LPR after LRYGB or OAGB between January 2017 and January 2022 were included in our database and retrospectively reviewed. All data pertaining to each patient, including demographic, preoperative, and postoperative clinical data, as well as data on previous bariatric surgery and indications for surgical revision, were collected. All patients included in the series underwent a 3 to 6 months program of dietary and psychological counseling to exclude altered eating behaviors or other dysfunctional psychological factors. The anatomic cause evaluation was performed by upper gastrointestinal series, upper gastrointestinal endoscopy, and three-dimensional gastric computed tomography with effervescent oral contrast medium (3D-CT gastric volumetry). The indications for LPR were weight regain ≥25% compared to the nadir weight achieved after OAGB or LRYGB in the presence of anatomical anomalies. For Nadir weight, we considered the lowest weight obtained by the patient among all available values regardless of the time of the postoperative period. Patients that experienced an insufficient weight loss after well-performed gastric bypass or that needed a lengthening of the intestinal loops were excluded from the present study. Patients with bypass-related complications (e.g., marginal ulcer and severe biliary reflux) who required a redo surgery for this reason were excluded. Two types of criteria were used to define the anatomical anomalies based on radiological and endoscopic findings. From a radiological point of view, a gastric pouch was considered dilated when its volume was calculated at >80 cm^3^ for LRYGB and >200 cm^3^ for OAGB upon a three-dimensional gastric computed tomography with an effervescent contrast medium and the reconstruction of gastric pouch. The endoscopic criterion considered both the diameter of the gastrojejunal anastomosis and the gastric pouch volume. All anastomoses >20 mm for LRYGB and >40 mm for OAGB were considered dilated, while a gastric pouch was considered endoscopically dilated when the retrovision maneuver with the gastroscope was easily performed. The selection criteria used in the present study were arbitrarily established on the basis of both our personal experience and data from similar studies published in the literature. All cases were discussed in a multidisciplinary meeting before the surgery.

Primary outcomes were weight loss after revisional surgery and resolution/improvement of comorbidities. Weight loss results were expressed as the change in body mass index (BMI) and percentage of total weight loss (%TWL). The remission of any comorbidity was defined as a patient that no longer needed drug therapy and those showing normal blood pressure and lab values. The resolution or improvement of type II diabetes mellitus (TIIDM) was defined by the discontinuation of diabetes medications with normal fasting blood glucose (<100 mg/dL or glycated hemoglobin < 6%) or a decreased dosage of diabetes medications. The postoperative resolution of obstructive sleep apnea syndrome (OSAS) was defined by the discontinuation of the use of continuous positive airway pressure. The secondary outcomes were the surgery’s duration and the rate of postoperative complications. Operating time was defined as the time between the first incision and the last stitch at skin level. Postoperative complications were classified based on the Clavien–Dindo classification (CD) [14] and the time of onset (early, during the first 30 days after surgery, or late, occurring after the first month). Continuous demographic variables and outcome variables are expressed as mean ± standard deviation. Categorical variables, in addition to complications, are reported as numbers and percentages. The comparison of continuous outcomes between the groups was carried out by means of a nonparametric unpaired test (i.e., Wilcoxon test). A 2-sided *p*-value < 0.05 was considered to be significant. Statistical analysis was performed using SAS JMP 10.0 software (SAS Institute Inc., Cary, NC, USA).

### 2.2. Surgical Technique

All procedures were performed laparoscopically with the 5-trocar technique (5–12 mm) and the patient in a modified lithotomy position. First, the length of the biliopancreatic loop in the OAGB and the alimentary and biliary loop in the LRYGB was measured to make sure the length was correct. We considered correct a length not inferior to 80 cm for the biliary loop and not inferior to 150 cm for the alimentary loop in the LRYGB. In the case of OAGB, the inferior limit of length for the biliary loop was estimated at 150 cm. Secondly, any residual adhesions, if present, were carefully excised, with particular attention to completely freeing the posterior wall of the gastric pouch and the alimentary limb. If a hiatal hernia was present, it was repaired directly by approaching the pillars with an interrupted nonabsorbable suture. Then, a 36 Fr-calibrating bougie was inserted orally by the anesthesiologist and laid out alongside the lesser curvature. The gastric pouch was then recalibrated using a linear mechanical stapler, removing the dilated lateral part of the pouch from the gastrojejunal anastomosis to the gastroesophageal junction (Figure 1a,b). When the gastrojejunal anastomosis was dilated, its recalibration was also performed using a linear mechanical stapler. The anastomosis was tested with methylene blue instilled through the orogastric bougie, which was pulled back at the end of the test. A drain was placed behind the gastrojejunal anastomosis inconstantly.

## 3. Results

During the period considered, 23 patients met the inclusion criteria. All patients were referred from other bariatric centers following WR after gastric bypass procedures. Demographic characteristics are shown in Table 1. Twelve patients had LRYGB, and 11 had OAGB. A previous bariatric surgery before the gastric bypass was present in 34.8% (8/22) of patients: four patients underwent sleeve gastrectomy and four had adjustable gastric banding (AGB). In particular, three patients underwent multiple bariatric surgeries: Two patients had gastric banding followed by sleeve gastrectomy and one patient had repeated two gastric banding before; the gastric bypass mean pre-bypass weight and mean pre-bypass BMI were 115.5 ± 19.7 kg and 43.3 ± 5.7 kg/m^2^, respectively; the mean weight and mean BMI at the time of resizing were 96.5 ± 13.9 kg and 36.3 ± 4.7 kg/m^2^, respectively (Table 2). Regarding pre-existing comorbidities, 30.4% (*n* = 7) had arterial hypertension, 17.4% (*n* = 4) had TIIDM, 17.4% (*n* = 4) had OSAS, and 8.7% (*n* = 2) had dyslipidemia.

All patients underwent resizing of the gastric pouch, and in three patents with previous LRYGB (3/23; 13%), it was also necessary to resize the GJA. Furthermore, hiatoplasty was associated with resizing of the gastric pouch in 6/23 cases (26.1%). The mean time between bypass and revision of the gastric pouch was 77.9 ± 54.5 months. The mean weight, mean BMI, and mean %TWL at 24.2 ± 16.1 months after LPR were, respectively, 77.9 ± 17.3 kg, 29.3 ± 5.8 kg/m^2^, and 19.6 ± 9%. The difference between the weight and BMI before resizing and the weight and BMI at the last follow-up visit was statistically significant (*p* = 0.004 and *p* = 0.00005). In five (21.7%) cases, the patient lost less than 10% of total weight. No statistically significant difference was found between LPR after LRYGB and OAGB (15.4 ± 7.4% vs. 22.5 ± 9.6%, *p* = 0.6). Given that after gastric bypass, the mean %TWL was 15.7 ± 7.9%, the overall mean %TWL after the two operations was 35.3 ± 13.1%.

Comorbidities had an overall resolution and/or improvement rate of 47%. In particular, in 17.6%, there was total resolution, and in 29.4% of cases, there was an improvement. Specifically, OSAS had complete resolution with the discontinuation of positive pressure therapy in 2/4 cases; in one patient, it improved; in another patient, it remained stable. Diabetes mellitus improved in two cases, leading to a reduction in medication, while the other patients remained stable. Hypertension resolved completely in one patient with discontinuation of therapy, three patients reduced the dosage of antihypertensives, and it remained stable in three cases.

There was no mortality in the present series, and the rate of conversion to laparotomy was nil. The mean operative time was 71.7 ± 21.9 min. Early postoperative complications occurred in two cases (8.7%), while no late complications were recorded. Both patients had LPR of a previous OAGB. The first patient developed hyperpyrexia with increased inflammatory markers at postoperative day (POD) 2. A CT scan was then performed, which did not reveal any anomalies. Empiric antibiotic therapy was administered, with a resolution of the hyperpyrexia and normalization of the inflammatory markers after 2 days (CDII). In the second case, the patient developed a fistula at the level of the cardia with a subsequent large abdominal collection. Therefore, the patient first underwent laparoscopic drainage of the collection (CDIIIb). In the following days, due to clinical worsening, she was reoperated on with a concomitant laparotomic and endoscopic approach. A thorough lavage of the abdominal cavity, the placement of two abdominal drains, and the endoscopic placement of a pigtail drainage were performed in the same operation. The fistula healed at POD 33, although it resulted in a stenosis of the cardia area, which was treated endoscopically with the placement of a stent.

## 4. Discussion

Due to its multifactorial origin, the approach to WR has to be multidisciplinary and systematic. In the majority of cases, the cause of WR can be found in the relapse of dietary and psychiatric aspects, associated with the maintenance of an incorrect lifestyle [6,15]. The correct psychological and nutritional preparation of the bariatric patient remains the best method for guaranteeing valid long-term results. Unfortunately, getting patients to understand the rapid and sometimes shocking changes in lifestyle habits and body image after bariatric surgery is sometimes very complicated, and consequently, the most fragile patients tend to fail and gain weight. For these reasons, if we want to decrease the risks of further failures, an even more meticulous psychological and nutritional assessment is of fundamental importance before thinking about a new surgical procedure. In some cases, however, weight gain is associated and/or facilitated by “anatomical causes” related to bariatric surgery. Notably, in the case of RYGB and OAGB, the generally accepted anatomical factors are gastric pouch dilatation and the enlargement of the gastrojejunal anastomosis (GJA) [6,16,17]. The dilation of the pouch and/or of GJA seems to be simultaneously supported by a physiological progressive dilation and possible incorrect eating behaviors (binge eating), which join to form a vicious circle [5,18]. Another possible cause is the poor quality of the surgery, especially in complicated patients with a high starting BMI. Previous gastric banding placement also seems to be involved. In fact, adhesions could make the preparation of the pouch more difficult, especially at the level of the posterior stomach wall in the cardia region. The incomplete release of this portion results in a primarily large pouch after gastric bypass. The physiological consequence of these anatomical anomalies would be the loss of the restrictive component and the lack of a sense of satiety, which occurs often early in the postoperative period [19].

The concept of WR after bariatric surgery presents a wide variety of definitions from the literature, resulting in a lack of standardization. Furthermore, for some period of time, the concept of WR has often been confused with that of insufficient weight loss. This confusion has further delayed the achievement of a common definition of WR. In a recent prospective study involving a cohort of more than 1400 patients, King et al. classified the weight regain using five continuous measures and eight different threshold measures. The authors found that at 5-year follow-up, according to different definitions, the percentage of patients with a diagnosis of WR ranged from 43.6% to 67.3% and the percentage of maximum weight lost had a stronger association with the recurrence of obesity-related complications and the deterioration of the quality of life [20]. Similarly, Voorwinde et al. prospectively analyzed 868 patients and found an even wider variability in WR (16–87%) based on the six different definitions used and concluded that identifying one single definition of WR related to clinical outcomes was difficult [21]. In the present study, we diagnosed WR after gastric bypass when the patient regained at least 25% of the lowest weight achieved in the postoperative period. In our daily practice, this measure is well-related to the worsening of the quality of life and to the recurrence of obesity-related comorbidities as diabetes. Indeed, without a correlation with the patient’s clinical condition, the WR measurement simply remains as a number with no real clinical implication. For this reason, the lack of the standardization of WR diagnoses risks distorting the real meaning of the long-term results of different surgical techniques for the obesity treatment.

Currently, there is no uniformly accepted definition concerning gastric pouch dilation, nor is there any standardization in its measurement [22]. Hamdi et al. in their series of 25 LRYGB patients for the definition of gastric pouch dilation used only endoscopic criteria based on not easy-to-standardize measures [23]. More recently, Ben Amour et al., by the use of tomography gastric volumetry, estimated a gastric pouch greater than 200 mL as dilated [24]. Regarding OAGB, the literature’s data are really scarce. Faul et al. evaluated a gastric pouch when the width was >4 cm as dilated upon a CT scan, and retrovision was possible upon gastroscopy [19]. Based on our experience with LRYGB and OAGB, we considered the respective values of 80 mL for LRYGB and 200 mL for OAGB as cut-off measures for dilation, evaluated by 3D-CT scan volumetry and in both cases confirmed by the possibility of easily performing the retrovision maneuver with the endoscope.

Concerning the stoma size, a 2 cm cut-off was chosen based upon two different criteria. The first, arbitrarily, related to our experience and the second related to literature data. In the last 10 years, we have progressively reduced the size of the anastomosis from 3–3.5 cm of the diameter to about 2 cm, recording a significant increase in the weight loss of patients during the first 24 postoperative months. Our unpublished and personal data seem to be confirmed also by several previous studies. Heneghan et al. in a very interesting study showed that a stoma diameter greater than 2 cm was independently associated with weight regain after RYGB [25]. Moreover, Abu Dayyeh et al. found that a stoma diameter greater than 2 cm correlated significantly with weight regain after RYGB [16]. The importance of stoma size in weight loss after LRGB was emphasized even more by the study by Ramos et al., which showed a highly significant difference in weight loss between patients with a 15 mm anastomosis and those with a 45 mm anastomosis [26]. For these reasons, we think that 2 cm is a balanced measure to evaluate a gastrojejunal anastomosis as effective with an acceptable risk of stenosis. To our knowledge, there are no studies correlating gastrojejunal anastomosis diameter and weight loss after OAGB.

In the presence of these anatomical anomalies, several surgical techniques have been proposed. The placement of a gastric banding at the level of the dilated gastric pouch appears to be effective in restoring the restrictive component, with encouraging weight loss results in the short and medium term [8,9]. Unfortunately, the complications traditionally described for this device, such as reflux, dysphagia, and slippage, have in several series led to a high rate of removal of the banding [12]. Vilallonga et al. proposed returning to normal anatomy with a sleeve gastrectomy in a single step as an option after WR [7]. However, the series is currently anecdotal, and even in highly experienced centers where this procedure is performed conventionally, the risk of postoperative complications remains high. The endoscopic alternatives, given the fewer complications and less invasiveness compared to surgical revisions, are rapidly increasing, with the advantage of having access to the area of interest without anatomical obstacles such as adhesions [27]. The use of the full-thickness suturing technique of the stoma associated or not with that of the gastric pouch with the use of OverStitch device (Apollo Endosurgery, Austin, TX, USA) seems to produce the best results even if conflicts are observed in different studies. Indeed, Thompson et al. demonstrated an efficacy and durability of the technique with an 8.8% TWL at 5 years of follow-up, while on the contrary, in the retrospective study by Callahan et al., the results at 5 years were significantly worse than those at one year after the procedure [27,28]. In different systematic reviews and meta-analyses, the average %TWL at one year varied between 5.8% and 8.6% [29,30,31]. Unfortunately, these are complex procedures requiring long learning curves, and the results seem to be limited and not long-lasting [32]. Finally, LPR with or without GJA resizing has been proposed, with contrasting results [19,24,33,34,35]. Ben Amor et al. reported the largest series to date with the longest follow-up, with 48 cases and 5 years of follow-up. In this study, they show 55% excess BMI loss (%EBMIL) 3 years after resizing. On the other hand, at 5 years, the results were less favorable, with 39% EBMIL [24]. Analogously, Borbély et al., in their series of contemporary gastric pouch reshaping and GJA resizing, recorded 43% EBMIL at 48 months [34]. Furthermore, Wijngaarden et al., in their comparative study of patients operated on with the resizing of the gastric pouch and GJA vs. the creation of a new GJA, recorded a %TWL slightly greater than 10% at 2 postoperative years, with no difference between the two techniques [35]. This difference in the results was also highlighted in our study. We contemporaneously resized the pouch and the GJA in only three patients, and we recorded a mean %TWL of 19%, but 21% of patients experienced a poor response, with a mean %TWL of 8% at 2 years. This high rate of poor results probably was in these patients once again linked to the multifactorial origins of WR and, in particular, to alimentary behavior errors.

It is important to note that the literature data on LPR after gastric bypass almost exclusively concern LRYGB. To date, only Faul et al. analyzed the outcomes of LPR for weight regain after OAGB, finding an impressive 31%TWL at 2 years of follow-up [19]. In our series, almost 50% of patients had previous OAGB, and the results were better than after LRYGB, even if the difference was not statistically significant. Although the malabsorptive component after OAGB plays a more important role than after LRYGB, the size of the gastric pouch is very important in determining satiety. In our experience, patients operated on for LPR after OAGB experienced a significantly greater reduction in food quantities and an earlier sense of satiety than patients operated on after LRYGB. This phenomenon, added to the malabsorption of the biliary loop, can justify the best results in the OAGB group.

Revisional bariatric surgery is burdened by a longer operating time and a higher complication rate than primary bariatric surgery, and this should be adequately considered before planning a surgical revision for WR [36]. A previous series on LPR and resizing of the GJA reported a complication rate ranging from 8% to 30% [19,24,33,34,37]. In the present study, an early postoperative complication occurred in two patients, with a rate of 8.7%; this is fully consistent with literature data. Of these two patients, only one needed a reintervention due to a staple line leak. The frequency of this type of complication is logically higher after LPR than after gastric bypass. Ben Amor et al. recorded three cases of staple line leaks and one case of intraabdominal abscess without an evident leak in their series of 48 LPR after LRYGB. This phenomenon probably has a two-fold cause. First, the extensive freeing of the gastric pouch and the GJA increases the risk of weakening the stomach wall and jeopardizing its vascularization. Secondly, the behavior of the surgeon who performs a surgery after the failure of a previous procedure could be more aggressive with the use of probes that are too small, thus significantly increasing the pressure inside the gastric pouch and decreasing the compliance of the gastric wall. Moreover, the complexity of revisional surgery is frequently combined with an increase in operative time [38]. In our experience, the operative time was slightly longer than in previous other series. This was probably due to the fact that 30% of patients had already had a previous bariatric surgery before the bypass. Furthermore, a quarter of patients needed hiatoplasty to treat an HH. In this situation, freeing the diaphragmatic pillars and the gastric pouch is certainly more surgically demanding and, consequently, more time-consuming.

The present study has several limitations. First, its retrospective nature and the small sample size are limitations. Second, the follow-up of only 2 years does not allow us to draw firm conclusions about the duration of the effects related to the restoration of gastric restrictions.

## 5. Conclusions

A large gastric pouch could cause the loss or at least a reduction in the restrictive effect resulting from the bypass and, consequently, could increase the risk of WR. In our series, LPR for WR showed good results in weight loss and in the improvement/resolution of comorbidities, with an acceptable complication rate and operative time. The few data in the literature and the complexity of the patients make this procedure feasible only in highly selected cases and in centers with adequate experience in the management of redo bariatric surgery. Only further studies with a greater cohort of patients and a longer postoperative follow-up will be able to highlight the long-term benefits of this technique.

## Figures and Tables

**Figure 1 jcm-11-06238-f001:**
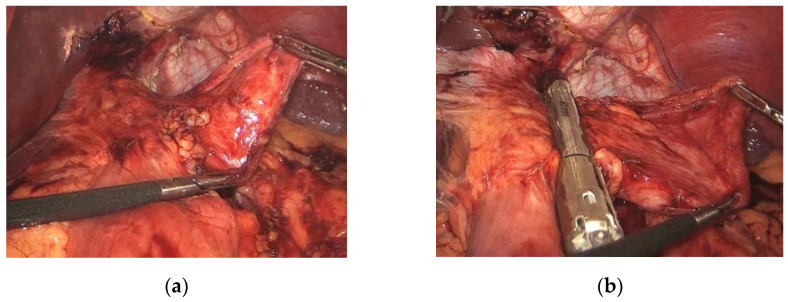
(**a**) Complete liberation of the dilated gastric pouch. (**b**) The gastric pouch was then recalibrated using a linear mechanical stapler, removing the dilated lateral part of the pouch from the gastrojejunal anastomosis to the gastroesophageal junction.

**Table 1 jcm-11-06238-t001:** Characteristics of the study population.

Age	47.1 ± 8.3 years
M/F	22/1
LRYGB/OAGB	12/11
BMI before bypass	43.3 ± 5.7 kg/m^2^
Weight before bypass	115.5 ± 19.7
Lower BMI after bypass	27.9 ± 6.2 kg/m^2^
BMI before resizing	36.3 ± 4.7 kg/m^2^
Weight before resizing	96.5 ± 13.9 kg
**Comorbidities** Hypertension Type II Diabetes Mellitus OSAS Dyslipidemia	30.4% 17.4% 17.4% 8.7%
Previous bariatric Surgery	34.8% (n.8)

LRYGB: laparoscopic Roux-en-Y gastric bypass; OAGB: one-anastomosis gastric bypass; BMI: body mass index; OSAS: obstructive sleep apnea syndrome.

**Table 2 jcm-11-06238-t002:** Peri-operative and postoperative outcomes.

Mean operative time	73.8 ± 21.6
Mean time between bypass and LPR (months)	77.9 ± 54.5
Mean follow up after LPR (months)	24.2± 16.1
Mean weight after LPR	77.9 ± 17.3 kg
Mean BMI after LPR	29.3 ± 5.8 kg/m^2^
Mean %TWL after LPR	19.6 ± 9%.

LPR: laparoscopic pouch resizing; BMI: body mass index; %TWL: % total weight loss.

## Data Availability

The data presented in this study are available upon request from the corresponding author (Dr. Sergio Carandina; sergio.carandina@gmail.com).

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
