# Peer review of "Resizing of the Gastric Pouch for Weight Regain after Laparoscopic Roux-en-Y Gastric Bypass and One-Anastomosis Gastric Bypass: Is It a Valid Option?"

_jcm, 2022, doi:10.3390/jcm11216238_

Round 1

Reviewer 1 Report

Well written paper, good results, relevant in clinical practice. However, your inclusion criteria are arbitrary. This is not bad, because there is currently no concensus about defining weight regain or indications for reinterventions. Please note clearly in Methods of both abstract and text that the criteria you use for this study were thought up by you and are not commonly used. Also remove the term prospectively (line 61), because your study is pure retrospective.

Author Response

Reviewer n.1

Well written paper, good results, relevant in clinical practice. However, your inclusion criteria are arbitrary. This is not bad, because there is currently no consensus about defining weight regain or indications for reinterventions. Please note clearly in Methods of both abstract and text that the criteria you use for this study were thought up by you and are not commonly used. Also remove the term prospectively (line 61), because your study is pure retrospective.

According to your suggestion, the term “prospectively” has been removed.

We thank the Reviewer for his constrictive feedback. 

Reviewer 2 Report

This study reports the outcome of pouch reshaping to address weight regain in 23 patients.

This paper is well written and of timely importance.

I only have a few questions and comments:

It is important to have a broad armamentarium in bariatric surgery as a life-long disease.

Please discuss the diagnosis of weight regain further, and also discuss the criteria for an enlarged pouch (volumetric cutoffs). Are there any data from apollo anastomotic resizing?

Please explain the follow-up of 2y, as patients from 2017 are included

Did you convert any patients from oagb to rygb due to weight regain?

Author Response

Reviewer n.2

This study reports the outcome of pouch reshaping to address weight regain in 23 patients.

This paper is well written and of timely importance.

I only have a few questions and comments:

It is important to have a broad armamentarium in bariatric surgery as a life-long disease.

Please discuss the diagnosis of weight regain further, and also discuss the criteria for an enlarged pouch (volumetric cutoffs). Are there any data from apollo anastomotic resizing?

 Response: We thanks the Reviewer for its positive comments. The concept of weight regain after bariatric surgery presents a wide variety of literature definitions resulting in a lack of standardization. Furthermore, for some time the concept of weight regain has often been confused with that of insufficient weight loss. This confusion has further delayed the achievement of a common definition of weight regain. In a recent prospective study involving a cohort of more than 1400 patients, King et al. classified the weight regain using 5 continuous measures and 8 different threshold measures. The authors found that at 5 years of follow-up, according to the different definitions the percentage of patients with a diagnosis of weight regain ranged from 43.6% to 67.3% and that the percentage of maximum weight lost had the stronger association with the recurrence of obesity related complications and deterioration of quality of life. Similarly, Voorwinde et al. prospectively analyzing 868 patients found an even wider variability of weight regain (16%-87%) based on the six different definitions used, and concluded that identifying one single definition of weight regain related to the clinical outcomes was difficult. In the present study, we diagnosed weight regain after gastric bypass when patient regained at least 25% of the lowest weight achieved in the postoperative period. In our daily practice this measure is well related to the worsening of the quality of life and to the recurrence of obesity-related comorbidities as diabetes after weight regain. Indeed, without a correlation with the patient's clinical condition, the weight regain measurement simply remains a number with no real clinical implication. For this reason, the lack of standardization of weight regain diagnosis risks to distort the real meaning of the long-term results of the different surgical techniques for the obesity treatment (This paragraph was added to the Discussion section).

King WC, Hinerman AS, Belle SH, Wahed AS, Courcoulas AP. Comparison of the performance of common measures of weight regain after bariatric surgery for association with clinical outcomes. JAMA; 320 (15): 1560-1569.

Voorwinde V, Steenhuis IHM, Janssen IMC, Montpellier VM, van Stralen MM. Definitions of long-term weight regain and the associations with clinical outcomes. Obes Surg 2020; 30: 527-536.

Regarding the volumetric cut-offs to define the enlarged gastric pouch, the lack of recommendation in the literature is even more evident. The problem is the lack of a reliable system for measuring the pouch volume. The few studies in the literature present either radiological or endoscopic measurement techniques or, as in our case, a combination of the two. But in any case, the cut-offs are arbitrarily established, based on the experience of the different authors. In a recent review, Mahawar et al. analyzed 14 different studies that correlated gastric pouch volume after RYGB with outcomes over time. In almost all studies, the measurement system was different and the data obtained heterogeneous. The authors identified a volume greater than 25 mm as indicative of worse results, but did not correlate it with weight regain. Furthermore, the data regarding the OAGB are even more scarce. In the Discussion section we compared our way to measure pouch volume to literature data. Unfortunately, the lack of postoperative data and the lack of standardization in pouch creation (boogie size, number of staplers, level of starting the gastric transversal section), prevents to obtain reliable cut-off measurements.

Mahawar K, Sharples AJ, Graham Y. A systematic review of the effect of gastric pouch and/or gastrojejunostomy (stoma) size on weight loss outcomes with Roux-en-Y gastric bypass. Surg Endosc 2020; 34: 1048-1060.

In recent years, the use of endoscopic techniques in the treatment of both obesity and weight regain after surgery has become increasingly popular. In particular, the Apollo system has made it possible to obtain results, in some cases encouraging. For this reason, on your advice, we have added the following paragraph to the discussion section: " The use of the of full-thickness suturing technique of the stoma associated or not with that of the gastric pouch with the use of OverStitch device (Apollo Endosurgery, Austin, TX) seems to give the best results even if they remain conflicting in the different studies. Indeed Thompson et al. demonstrated an efficacy and durability of the technique with an 8.8% TWL at 5 years of follow-up, while on the contrary, in the retrospective study by Callahan et al. the results at 5 years were significantly worse than those at one year after the procedure. In different systematic reviews and meta-analysis the average %TWL at one year varied between 5.8% and 8.6%.”

Thompson CC, Chand B, Chen YK, DeMarco DC, Miller L, Schweitzer M, Rothstein RI, Lautz DB, Slattery J, Ryan MB, Brethauer S, Schauer P, Mitchell MC, Starpoli A, Haber GB, Catalano MF, Edmundowicz S, Fagnant AM, Kaplan LM, Roslin MS. Endoscopic suturing for transoral outlet reduction increases weight loss after Roux-en-Y gastric bypass surgery. Gastroenterology 2013; 145: 129-137.

Callahan ZM, Su B, Kuchta K, Linn J, Carbray J, Ujiki M. Five-year results of endoscopic gastrojejunostomy revision (transoral outlet reduction) for weight gain after gastric bypass. Surg Endosc 2020; 34: 2164-2171

Dhindsa BS, Saghir SM, Naga Y, Dhaliwal A, Ramai D, Cross C, Singh S, Bhat I, Adler DG. Efficacy of transoral outlet reduction in Roux-en-Y gastric bypass patients to promote weight loss: a systematic review and meta-analysis. Endosc Int Open 2020; 8: E1332-E1340

Jaruvongvanich V, Vantanasiri K, Laoveeravat P, Matar RH, Vargas EJ, Maselli DB, Alkhatry M, Fayad L, Kumbhari V, Fittipaldi-Fernandez RJ, Hollenbach M, Watson RR, Gustavo de Quadros L, Galvao Neto M, Aepli P, Staudenmann D, Brunaldi VO, Storm AC, Martin JA, Gomez V, Abu Dayyeh BK. Endoscopic full-thickness suturing plus argon plasma mucosal coagulation versus argon plasma mucosal coagulation alone for weight regain after gastric bypass: a systematic review and meta-analysis. Gastrointest Endosc 2020; 92: 1164-1175.

Brunaldi VO, Jirapinyo P, de Moura DTH, Okazaki O, Bernardo WM, Galvão Neto M, Campos JM, Santo MA, de Moura EGH. Endoscopic Treatment of Weight Regain Following Roux-en-Y Gastric Bypass: a Systematic Review and Meta-analysis. Obes Surg 2018; 28: 266-276

Please explain the follow-up of 2y, as patients from 2017 are included

Response: The majority of patients were operated on in 2019, 2020 and 2021. Only 4 patients were operated on in 2017 and 2018. For these reasons, the average follow-up is only 24 months, although some patients had more than 4 years follow-up.

Did you convert any patients from oagb to rygb due to weight regain?

Response: Yes, we did. In some patients, not included in the present study, that presented not responsive biliary reflux associated to weight regain we converted OAGB into RYGB. Unfortunately, in our experience, this operation worked very well in treating biliary reflux but it was unsuccessful in obtaining a significant weight loss.

We thank again the Reviewer for his constructive feedback. 

Reviewer 3 Report

This paper collects the experience, in a complex issue to management of weight regain due to reservoir and anastomotic dilation. So, congratutations to the authors.

The Methology and Results are widely described, with adequate follow-up at 2 years.

In the discussion,  it would be ideal to know  what percentage of subjects undergoing gastric bypass and OAGB could present dilatation of the reservoir and/or the anastomosis over time (i.e. 5 years) that were susceptible to different procedures (endoscopic and/or surgery).

 Second, at what point should you consider whether a patient may have a dilated reservoir? From what percentage of weight regain? 

Author Response

Reviewer n.3

This paper collects the experience, in a complex issue to management of weight regain due to reservoir and anastomotic dilation. So, congratutations to the authors.

The Methology and Results are widely described, with adequate follow-up at 2 years.

In the discussion,  it would be ideal to know  what percentage of subjects undergoing gastric bypass and OAGB could present dilatation of the reservoir and/or the anastomosis over time (i.e. 5 years) that were susceptible to different procedures (endoscopic and/or surgery).

Response: We thanks the Reviewer for its positive comments. We agree with the reviewer that having these data would be ideal for evaluating the evolution of different procedures over time. Unfortunately, there are no data in the literature that correlate WR with long-term gastric pouch and/or stoma dilation. In the different study it is pointed out that 20-30% of patients operated on for RYGB have a long-term failure of the procedure. For the OAGB, the data are even more scarce but we are probably on the same order of figures. However, this percentage of patients includes all patients regardless of the underlying cause. For this reason, the diagnosis of WR due to anatomical anomalies is often made by exclusion. Before proposing revisional surgery to these patients, we start a multidisciplinary evaluation including psychological and nutritional protocol with frequent clinic visits. Only in the event that the patient does not show a significant improvement, we consider the anatomical causes as pivotal in the process of WR.

Second, at what point should you consider whether a patient may have a dilated reservoir? From what percentage of weight regain? 

Response: The concept of weight regain after bariatric surgery presents a wide variety of literature definitions resulting in a lack of standardization. Furthermore, for some time the concept of weight regain has often been confused with that of insufficient weight loss. In the present study, we diagnosed weight regain after gastric bypass when patient regained at least 25% of the lowest weight achieved in the postoperative period.  But the most important thing is the change in satiety. An increase in the food quantity long-term after the bypass is physiological, but when this increase becomes too important and above all the patient reports the loss of the sense of satiety, then the suspicion of the presence of anatomical anomalies is necessary. From a purely anatomical point of view, there is a lack of agreement regarding the volumetric cut-offs to define a gastric pouch enlarged. The selection criteria used in our study were arbitrarily established on the basis of both our personal experience in gastric pouch construction during bypass and data from similar studies published in the literature.

We thank again the Reviewer for his positive and constructive feedback. 

Round 2

Reviewer 2 Report

thank you for your revisions. I think they significantly improved the manuscript